# AI-Hamilton: Leveraging In-Context Learning for Modeling Hamiltonian Systems

## Abstract

We present a novel approach to learning Hamiltonian systems from observational data, combining the strengths of in-context learning (ICL) and hypernetworks with the rigorous guarantees of structure-preserving numerical methods. ICL, a unique and powerful capability exhibited by large language models (LLMs), enables pre-trained LLMs to adapt their predictions based on auxiliary information known as "context". While a few studies have explored applying ICL to neural operator learning, most of these approaches treat operators as "black-boxes," offering no guarantees of physical consistency in their predictions. To address this limitation, we propose ICL-based neural operators explicitly designed to preserve the symplectic structure inherent to Hamiltonian dynamical systems. Through extensive experiments on a range of Hamiltonian systems, we demonstrate the proposed model's ability to maintain structural fidelity while achieving improved prediction accuracy compared to black-box ICL-based operators.

## 1 Introduction

Complex physical processes are often described by systems of differential equations whose modeling plays a crucial role in science and engineering applications. Enabled by the success of deep learning, data-driven approaches to such modeling have seen significant advancements, engendering the relatively new discipline of scientific machine learning (SciML) (Baker et al., 2019; Karniadakis et al., 2021). In cases where large datasets are available and governing equations are unknown, neural operators (NOs) (Azizzadenesheli et al., 2024; Lu et al., 2021; Li et al., 2021) have emerged as a promising method for inferring underlying operators in complex physics problems, leading to the development of various NO architectures (Tran et al., 2023; Li et al., 2023; Hao et al., 2023). However, most NOs function as sophisticated regression models that struggle to adapt to new "contexts," e.g., initial conditions far from those encountered during training.

Conversely, large language models (LLMs) (Brown et al., 2020; Achiam et al., 2023; Touvron et al., 2023), particularly those based on Transformer architectures (Vaswani et al., 2017), demonstrate a unique capability known as in-context learning (ICL) (Brown et al., 2020; Xie et al., 2022; Von Oswald et al., 2023). These models can perform better on specific tasks when prompted online with input/output examples of that task, effectively learning and adjusting their behavior without requiring explicit parameter updates (Von Oswald et al., 2023). Inspired by this capability, the design of neural operators has been informed with LLM-like architectures (Yang et al., 2023; Liu et al., 2024; Serrano et al., 2024; Kang et al.), and these models have shown improved performance, particularly in parametric and temporal extrapolation settings. However, despite these advances, existing NOs, including those leveraging ICL, primarily function as "black-boxes". In particular, there are no mechanisms in place to ensure physical consistency in their predictions, meaning that critical conservation laws and involution constraints necessary for dynamical stability are often ignored at prediction time.

Thankfully, considerable progress has been made on another front of SciML in developing models that guarantee physically consistent outputs (Greydanus et al., 2019). Unlike physics-informed approaches (Raissi et al., 2019), which impose physical consistency weakly via soft constraints, these methods enforce physical laws exactly through the design of neural network architectures that reflect geometric and topological structure (Greydanus et al., 2019; Cranmer et al., 2020; Toth et al., 2019; Lee et al., 2021) or leverage invariance/equivariance principles (Battaglia et al., 2018; Satorras et al., 2021). Of particular relevance are Hamiltonian neural networks (HNNs) (Greydanus et al., 2019) and their variants (Finzi et al., 2020; Chen et al., 2021), which incorporate concepts like symplecticity into model design to strictly satisfy consequences of Hamilton's least-action principle, including the law of energy conservation.

In this study, we aim to improve the utility of Hamiltonian dynamics learning through the development of a model that combines ICL capabilities with the ability to predict trajectories of energy-conserving systems. To this end, we propose an encoder-decoder-type Transformer model to approximate the system state, where the encoder processes example

trajectories to extract context which the decoder can use to adjust its predictions for new queries. However, instead of simply allowing the Transformer's output to serve as the predicted state trajectory over time (which is unlikely conserve energy), we use a hypernetwork (Ha et al., 2017) trained on the decoder's hidden representations to perform in-context modulation of the weights and biases of an auxiliary HNN. Since the decoder acts as a "black-box" neural operator, we speculate that utilizing its internal operational representation, encoded in its hidden neurons, enables the computation of effective and generalizable online-updates to an HNN based on context contained in state mini-trajectories. Moreover, this HNN can be trained alongside both the Transformer and hypernetwork using a database of state trajectories at different initial conditions, corresponding to Hamiltonians at parameter instances, to provide a prediction engine for parameterized Hamiltonian system states. In contrast to the predictions generated directly by the Transformer, this procedure has a distinct advantage: the trajectories produced by simulating the symplectic gradient flow of the learned HNN obey the laws of energy conservation and symplecticity, as required by Hamilton's least-action principle.

Our contributions include:

- development of an ICL-based meta-learning framework for learning a forecasting operator that is guided by Hamilton's least-action principle,
- design of a hypernetwork inferring a Hamiltonian system without requiring *a priori* knowledge on system parameters, and
- comprehensive experimentation on nine benchmark Hamiltonian systems that exhibit diverse characteristics (e.g., low-dimensional—1D/2D, high-dimensional—3D/5D/7D, chaotic vs. non-chaotic).

## 2 PRELIMINARIES

To provide context for our approach, we begin by reviewing two key technical preliminaries regarding neural operators and principles of structure preservation in Hamiltonian systems.

### 2.1 LEARNING OPERATORS

Neural operators (NOs) are a class of data-driven surrogate models that learn mappings between functions rather than input/output data, making them well-suited for solving problems governed by partial differential equations (PDEs). NOs take in discrete representations of continuous functions as input and output queryable network models, enabling them to serve as effective surrogates for complex physical systems. In this study, we focus on NOs as a forecasting mechanism, mapping a function representing past state evolution to a function representing future state evolution. In the following, we formally introduce NOs, outline the ICL paradigm for operator learning, and then describe our specific approach to learning forecasting operators.

**Neural operators** Consider two function spaces $\mathcal{X}, \mathcal{Y}$ and a nonlinear operator $\mathcal{A} : \mathcal{X} \to \mathcal{Y}$ mapping between them. Neural operators represent a family of machine learning methods developed for constructing a surrogate $\hat{\mathcal{A}} \approx \mathcal{A}$ lying in a trial space of neural networks. Under the assumption of access to input/output pairs $(x^i, y^i)_{i=1}^n$ such that $x^i \in \mathcal{X}$ and $y^i = \mathcal{A}(x^i) \in \mathcal{Y}$, a neural operator can be trained in the standard way, i.e., through solving the optimization problem:

$$\min_{\Theta} \sum_{i=1}^{n} L\left(\hat{\mathcal{A}}(x^i), y^i\right),$$

where $L$ denotes a certain discrepancy measure and $\Theta$ denotes the parameters of the neural operator $\hat{\mathcal{A}}$. In practice, the functional input data $x^i$ are first discretized; for example, if $x(t)$ and $y(t)$ are time-dependent scalar functions, the grid representation $\boldsymbol{x} = [x(t_1), \ldots, x(t_m)]^\mathsf{T} \in \mathbb{R}^m$ serves as input along with an evaluation point (or set of points) $t_p$ in the domain of $x$, and the neural operator learns a mapping $\boldsymbol{x} \mapsto \hat{\mathcal{A}}(\boldsymbol{x})$ that predicts $\hat{\mathcal{A}}(\boldsymbol{x})(t_p) \approx y(t_p)$ (Boullé & Townsend, 2023; Patel et al., 2021; Lu et al., 2021; Li et al., 2021). It will be convenient to assume that the locations in time where $\hat{\mathcal{A}}(\boldsymbol{x})$ is queried are identical to the locations $\boldsymbol{t} = [t_1, \ldots, t_m]$ where $x(t)$ is discretized, so that $(\boldsymbol{t}, \boldsymbol{x}) \mapsto \boldsymbol{y}$ (under $\mathcal{A}$) produces a vector $\boldsymbol{y}$ of operator evaluations, and therefore samples $(\boldsymbol{t}, \boldsymbol{x}, \boldsymbol{y})$ form the training data for the NO learning problem.

**In-context operator learning** Among the general class of neural operators just described, there is a particular type of neural operator that mimics the ICL property of LLMs (Yang et al., 2023; Liu et al., 2024; Serrano et al., 2024; Kang et al.). These models are designed as large-scale Transformers and trained on many diverse input/output pairs. That is, they are designed to optimize the model's output by taking in a set of example input/output pairs generated from a system of interest, analogous to the "prompt" in LLMs. For example, consider a learning problem where the training data for the unknown operator to determine is separated into $j$ groups; we assume that there are $n_j$ number of

input/output pairs in the $j$-th training dataset $\mathcal{D}_j = \left\{ (\boldsymbol{x}_j^i, \boldsymbol{y}_j^i) \right\}_{i=1}^{n_j}$. This arises, for example, when the operator to learn is parameterized and data is collected at various parameter instances indexed by $j$. Then, the input to the Transformer model can contain additional context which we call the prompt,

$$\boldsymbol{Z}_j = \left[ \underbrace{\begin{array}{ccc} \boldsymbol{x}_j^1 & \cdots & \boldsymbol{x}_j^{n_j-1} \\ \boldsymbol{y}_j^1 & \cdots & \boldsymbol{y}_j^{n_j-1} \end{array}}_{\text{prompt}} \middle| \begin{array}{c} \boldsymbol{x}_j^{n_j} \\ \boldsymbol{0} \end{array} \right]. \tag{1}$$

Given this input containing $n_j - 1$ examples comprising the prompt and a query input $\boldsymbol{x}_j^{n_j}$, the Transformer model is expected to produce a prediction $\tilde{\boldsymbol{y}}_j^{n_j} = \hat{\mathcal{A}}(\boldsymbol{t}_j, \boldsymbol{Z}_j) \approx \boldsymbol{y}_j^{n_j}$ which approximates the desired target.

**Learning a forecasting operator** The specific goal targeted in this study involves the prediction of future dynamical system states given a set of measurements on past system states. As an example, consider any homogeneous ODE $\frac{\mathrm{d}u}{\mathrm{d}t} = f(u)$ and an associated initial condition $u(0) = u_0 \in \mathbb{R}^m$. The forecasting operator (a.k.a., forecaster) serves as a surrogate for the flow map $\varphi_t(u_0) = u(t)$ which integrates this ODE system starting at $u_0$, i.e., the forecaster learns to predict future states of the system $u(t)|_{T<t<2T}$ given past measurements $u(t)|_{0<t<T}$ of the same system (from trajectories with varied initial conditions). Note that this forecasting operator is not instantaneous like the continuous flow map, but is informed with additional temporal context which may come from the same trajectory (passing through $u_0$) or from a combination of different trajectories.

## 2.2 LEARNING STRUCTURE-PRESERVING AND CONSERVATION-AWARE DYNAMICS

Structure-aware approaches to neural dynamics modeling aim to respect the physical laws governing a system, either by embedding known structure or by discovering conserved quantities from data. To this end, this study considers largely two distinct approaches: a HNN-style method that enforces the Hamiltonian structure preservation, and a Noether-network-style method that enforces invariance of conserved quantities. Structure preserving methods incorporate known geometric or topological information about the system at hand, such as symplecticity in the case of Hamiltonian systems. These models ensure that the learned dynamics are consistent with the underlying physics, providing both improved prediction stability and out-of-distribution generalizability. This study focuses specifically on energy-conserving systems, utilizing Hamiltonian neural networks for structure preservation. Noether networks (Alet et al., 2021), on the other hand, aim to discover conserved quantities directly from trajectories by enforcing time invariance, without requiring prior knowledge of the underlying physics.

**Hamiltonian systems and HNNs** The state of a (canonical) Hamiltonian system is typically described by a set of generalized coordinates and their associated momenta, denoted as $\boldsymbol{q} = [q_1, ..., q_n]$ and $\boldsymbol{p} = [p_1, ..., p_n]$, respectively. The integer $n$ represents the dimension of the system, which corresponds to $2n$ independent functions (i.e., degrees of freedom) required to uniquely specify the configuration of the system. The dynamics of a Hamiltonian system take place in the phase space, which is the product space of $\boldsymbol{q}$ and $\boldsymbol{p}$, yielding antisymmetric equations of motion. Particularly, these dynamics are described by the Hamiltonian function $\mathcal{H}(\boldsymbol{q}, \boldsymbol{p})$, representing the total energy, i.e., the sum of the kinetic and potential energies. This leads to the expressions

$$\frac{\mathrm{d}\boldsymbol{q}}{\mathrm{d}t} = \frac{\partial \mathcal{H}}{\partial \boldsymbol{p}}, \quad \frac{\mathrm{d}\boldsymbol{p}}{\mathrm{d}t} = -\frac{\partial \mathcal{H}}{\partial \boldsymbol{q}}, \tag{2}$$

describing the equations of motion through the symplectic gradient of the Hamiltonian, i.e., $[\dot{\boldsymbol{q}}, \dot{\boldsymbol{p}}]^\intercal = \operatorname{sgrad} \mathcal{H}(\boldsymbol{q}, \boldsymbol{p})$ (Marsden & Ratiu, 1998). Following this symplectic gradient described in Eq. (2), the Hamiltonian system is guaranteed to conserve the total energy along solution trajectories, as confirmed by the vanishing of the dissipation rate $\frac{\mathrm{d}\mathcal{H}}{\mathrm{d}t} = \left(\frac{\partial \mathcal{H}}{\partial \boldsymbol{q}}\right)^\intercal \frac{\mathrm{d}\boldsymbol{q}}{\mathrm{d}t} + \left(\frac{\partial \mathcal{H}}{\partial \boldsymbol{p}}\right)^\intercal \frac{\mathrm{d}\boldsymbol{p}}{\mathrm{d}t} = \left(\frac{\partial \mathcal{H}}{\partial \boldsymbol{q}}\right)^\intercal \frac{\partial \mathcal{H}}{\partial \boldsymbol{p}} - \left(\frac{\partial \mathcal{H}}{\partial \boldsymbol{p}}\right)^\intercal \frac{\partial \mathcal{H}}{\partial \boldsymbol{q}} = 0.$

HNNs approximate Hamiltonian systems by parameterizing the Hamiltonian function as a neural network $\mathcal{H}_\Theta(\boldsymbol{q}, \boldsymbol{p})$ with a set of learnable model parameters $\Theta$. HNNs are trained by minimizing a loss consistent with the equations of motion,

$$\mathcal{L}_{\text{symp}} = \left\| \frac{\mathrm{d}\boldsymbol{q}}{\mathrm{d}t} - \frac{\partial \mathcal{H}_\Theta}{\partial \boldsymbol{p}} \right\|_2^2 + \left\| \frac{\mathrm{d}\boldsymbol{p}}{\mathrm{d}t} + \frac{\partial \mathcal{H}_\Theta}{\partial \boldsymbol{q}} \right\|_2^2, \tag{3}$$

which we denote by the "symplecticity loss".

Figure 1: An overview of the proposed architecture. An encoder-decoder transformer processes a prompt, extracts context, and predicts future states given a novel query. A hypernetwork takes the decoder's hidden representations and infers a subset of model parameters inside an HNN. This HNN is then used to align the decoded predictions with Hamilton's least-action principle.

**Noether networks** Noether networks are neural models that discover conserved quantities in dynamical systems from data. Inspired by Noether's theorem, they learn a function representing a candidate conserved quantity and enforce its time-invariance along predicted trajectories. Unlike Hamiltonian or Lagrangian approaches, they do not require prior knowledge of the system's physics. A separate network $g_\phi$ is trained to represent the conserved quantity, typically by minimizing the Noether loss $\mathcal{L}_{\text{Noether}} = \sum_{t=1}^{T} \|g_\phi(\tilde{x}_t) - g_\phi(\tilde{x}_{t-1})\|$. At test time, $g_\phi$ and the loss are evaluated and minimized on predicted trajectories to enforce conservation.

## 3 METHODOLOGY

We now introduce a novel NO framework, AI-Hamilton, which integrates ICL-based operator learning with structure-preserving modeling techniques to ensure that future state predictions remain consistent with the underlying Hamiltonian physics. AI-Hamilton is built on an encoder-decoder Transformer model, $\boldsymbol{f}^{\text{TF}}$, where the encoder processes example trajectories $\mathcal{D}$ (the "prompt") and extracts context, while the decoder, conditioned on this extracted context and the target temporal locations, predicts future states given history information about a new trajectory. AI-Hamilton further employs a hyper-network to handle structure preservation, which takes hidden representations of the decoder as input and outputs the model parameters of an HNN. Finally, AI-Hamilton uses this inferred HNN to align the decoder's predictions with Hamilton's least-action principle, hence guaranteeing energy conservation and symplecticity in the learned dynamics. Figure 1 summarizes the computational flow of the proposed AI-Hamilton architecture.

### 3.1 INPUT-TO-OUTPUT PIPELINE

We begin with a detailed description of the overall AI-Hamilton pipeline. The input to AI-Hamilton adopts the standard format described in Eq. (1), with slight modifications. For notational simplicity, we define $\boldsymbol{u}_i = [\boldsymbol{q}(t_i), \boldsymbol{p}(t_i)] \in \mathbb{R}^{2n}$, where $\boldsymbol{q}(t_i) = [q(t_i), \ldots, q_n(t_i)]$ and $\boldsymbol{p}(t_i) = [p_1(t_i), \ldots, p_n(t_i)]$ denote the state variables measured at time $t_i$.

Similarly to other NOs, the Transformer operator $\boldsymbol{f}^{\text{TF}}(\boldsymbol{t}, \boldsymbol{Z}) = \boldsymbol{f}_d^{\text{TF}}(\boldsymbol{t}, \boldsymbol{f}_e^{\text{TF}}(\boldsymbol{Z}))$ at the first step of AI-Hamilton is comprised of an encoder $\boldsymbol{f}_e^{\text{TF}}$, which takes a prompt $\boldsymbol{Z}$ as input, and a decoder $\boldsymbol{f}_d^{\text{TF}}$ which processes the encoded input and a vector of temporal locations $\boldsymbol{t}$ at which to evaluate the NO. To facilitate the causal nature of dynamical trajectories, the prompt $\boldsymbol{Z}$ is augmented with two additional pieces of information besides the state variable $\boldsymbol{u}$: temporal indices and an example index. In particular, for the $i$-th example,

$$\boldsymbol{x}^i = \begin{bmatrix} t_1^i & \cdots & t_T^i \\ \boldsymbol{u}_1^i & \cdots & \boldsymbol{u}_T^i \\ \boldsymbol{e}_i & \cdots & \boldsymbol{e}_i \end{bmatrix}, \quad \boldsymbol{y}^i = \begin{bmatrix} t_{T+1}^i & \cdots & t_{2T}^i \\ \boldsymbol{u}_{T+1}^i & \cdots & \boldsymbol{u}_{2T}^i \\ -\boldsymbol{e}_i & \cdots & -\boldsymbol{e}_i \end{bmatrix},$$

where $\boldsymbol{e}_i$ is a canonical one-hot vector (of dimension $\max_i\{n_i\}$) with one in the $i$-th position and zeros otherwise. Following Yang et al. (2023), this one-hot vector is included to provide causal information about the temporal location of the state. With this definition of an example pair, we now define the $j$-th input identically to (1). Taking into account the goal of predicting the dynamics $T$ steps ahead, the Transformer simply processes this input and the time indices $\boldsymbol{t}^{n_j} = [t_{T+1}, \ldots, t_{2T}]$. This yields the desired state prediction through $\tilde{\boldsymbol{y}}_j^{n_j} = \boldsymbol{f}^{\mathrm{TF}}(\boldsymbol{t}^{n_j}, \boldsymbol{Z}_j)$.

Referring to Figure 1, the next step of AI-Hamilton involves the decoding extractor $\boldsymbol{f}^e$, which concentrates the dynamics information contained in the decoder's hidden representations into a latent vector $\boldsymbol{z}$. Formally, this step can be considered as an operation $\boldsymbol{z} = \boldsymbol{f}^e \circ \boldsymbol{f}^{\mathrm{TF}}(\boldsymbol{t}, \boldsymbol{Z})$ directly on the Transformer NO (see section 3.3 for a precise description). The resulting latent vector $\boldsymbol{z}$, representing a distillation of the decoder's world model, then serves as input to the hypernetwork $\boldsymbol{f}^{\mathrm{hyper}}$ which performs modulation. In particular, $\boldsymbol{f}^{\mathrm{hyper}}$ is designed to modulate the HNN $\mathcal{H}_\Theta$, which forms the potential function for AI-Hamilton's dynamics. Said differently, the output $\Theta_{\mathrm{indv}} = \boldsymbol{f}^{\mathrm{hyper}}(\boldsymbol{z})$ provides an online update to the HNN's base parameters $\Theta_{\mathrm{base}}$ (c.f. section 3.3), ensuring that the context contained in the Transformer input $\boldsymbol{Z}$ has a downstream effect on the learned Hamiltonian function $\mathcal{H}_\Theta$. Finally, the symplectic gradient of the HNN $\mathcal{H}_\Theta$ is computed at the state associated to the query in $\boldsymbol{Z}$, and structure-preserving dynamics are generated through the integral curves of this field. The remainder of this section provides further detail regarding each step of this process.

## 3.2 TRANSFORMERS: BASE MODELS FOR ICL

As in Yang et al. (2023), we choose a standard encoder-decoder transformer model consisting of softmax-based multi-head self-attention blocks (MHSAs) and cross-attention blocks (MHCAs) (Vaswani et al., 2017). The encoder processes a prompt by employing a series of MHSA layers and the decoder processes a target query (i.e., temporal history information from a novel trajectory) via a series of MHCA layers. We denote the hidden representations of the $\ell$-th encoder and decoder layer by $\boldsymbol{h}_i^\ell = (\boldsymbol{h}_{i,1}^\ell, \ldots, \boldsymbol{h}_{i,T_i}^\ell) \in \mathbb{R}^{d \times T_i}, \ell = 1, \ldots, L_i, i = \{\mathrm{enc}, \mathrm{dec}\}$, where $T_i$ denotes the number of tokens, and $L_i$ denotes the number of layers.

**Multi-head attention** We employ the softmax-based standard attention mechanism. Given query, key, and value matrices, $\boldsymbol{Q} \in \mathbb{R}^{T_q \times d}$, $\boldsymbol{K} \in \mathbb{R}^{T_k \times d}$, and $\boldsymbol{V} \in \mathbb{R}^{T_k \times d}$,[1] the attention mechanism is defined as $\mathrm{Att}(\boldsymbol{Q}, \boldsymbol{K}, \boldsymbol{V}) = \mathrm{softmax}\left(\frac{\boldsymbol{Q}\boldsymbol{K}^\top}{\sqrt{d_h}}\right)\boldsymbol{V}$, where the the softmax function is applied row-wise. The multi-head attention mechanism operating on $h$ heads can be defined as $\mathrm{MHA}(\boldsymbol{Q}, \boldsymbol{K}, \boldsymbol{V}) = \mathrm{Concat}(\boldsymbol{O}_1, \ldots, \boldsymbol{O}_h)\boldsymbol{W}^O$, where $\boldsymbol{W}^O \in \mathbb{R}^{d \times d}$ is a learnable matrix and $\boldsymbol{O}_i = \mathrm{Att}(\boldsymbol{Q}\boldsymbol{W}_i^Q, \boldsymbol{K}\boldsymbol{W}_i^K, \boldsymbol{V}\boldsymbol{W}_i^V)$ is defined in terms of learnable matrices $\boldsymbol{W}_i^Q, \boldsymbol{W}_i^K, \boldsymbol{W}_i^V \in \mathbb{R}^{d \times d/h}$. Setting $\boldsymbol{Q} = \boldsymbol{K} = \boldsymbol{V} = \boldsymbol{X}$ yields the self-attention mechanism (which has the same inputs for queries, keys, and values) while setting $\boldsymbol{Q} = \boldsymbol{X}$ and $\boldsymbol{K} = \boldsymbol{V} = \boldsymbol{Y}$ yields the cross-attention mechanism.

**Transformer layers** Each layer in the encoder and decoder follows the same structural pattern: multi-head attention (MHA), followed by residual connection, feed-forward network propagation (FF), and layer normalization (LN). The $\ell$-th Transformer layer can be written as

$$\tilde{\boldsymbol{h}}^{\ell-1} = \mathrm{LN}(\boldsymbol{h}^{\ell-1} + \mathrm{MHA}(\boldsymbol{h}^{\ell-1}, \boldsymbol{k}, \boldsymbol{v})),$$

$$\boldsymbol{h}^\ell = \mathrm{LN}(\tilde{\boldsymbol{h}}^{\ell-1} + \mathrm{FF}(\tilde{\boldsymbol{h}}^{\ell-1})),$$

where the feed-forward network is applied in a token-wise fashion. In the encoder, we set the input equal to the prompt, and $\boldsymbol{k} = \boldsymbol{v} = \boldsymbol{h}_{\mathrm{enc}}^{\ell-1}$ for all layers (i.e., self-attention). Conversely, for the decoder, we set the decoder tokens $\boldsymbol{h}_{\mathrm{dec}}$ as queries, and keys/values are the encoder outputs $\boldsymbol{h}_{\mathrm{enc}}$ used for cross-attention.

## 3.3 HYPERNETWORK: FROM DECODER TO HNNS

In this subsection, we explain how to obtain the latent code $\boldsymbol{z}$ from the decoder and infer the subset $\Theta_{\mathrm{indv}}$ of model parameters $\Theta_{\mathrm{HNN}} = \Theta_{\mathrm{base}} \cup \Theta_{\mathrm{indv}}$ defining the Hamiltonian function. The hypernetwork $\boldsymbol{f}^{\mathrm{hyper}}$ is defined as an $L$-layer perceptron $\boldsymbol{z} \mapsto \Theta_{\mathrm{indv}}$ with $\tanh$ activation function. The following provides details about the hypernetwork, including the production of its input, its processing mechanism, and its resulting use in conjunction with HNNs.

**Input: Decoder representation extractor** As mentioned, AI-Hamilton employs an extractor $\boldsymbol{f}^e$ that infers a latent code, denoted as $\boldsymbol{z} := \boldsymbol{f}^e(\boldsymbol{h}_{\mathrm{enc}}^{L_{\mathrm{enc}}}, [\boldsymbol{h}_{\mathrm{dec}}^1, \ldots, \boldsymbol{h}_{\mathrm{dec}}^L])$. The extractor processes the hidden representations of internal tokens from each decoder layer, conditioned on the encoder context $\boldsymbol{h}_{\mathrm{enc}}^{L_{\mathrm{enc}}} \in \mathbb{R}^{d \times T_{\mathrm{enc}}}$. To compress decoder information while preserving cross-layer dynamics, we parameterize $\boldsymbol{f}^e$ using a convolutional-layer-based architecture.

---

[1]For simplicity of exposition, we assume that the query/key and value dimensions are the same, i.e., $d = d_k = d_v$.

The latent code $\boldsymbol{z} = [\boldsymbol{z}_{\text{enc}}, \boldsymbol{z}_{\text{dec}}]$ involves two quantities extracted via $\boldsymbol{f}^e$ in separate stages. The first stage applies a mean aggregator to the encoder's final token representations. That is, we take the Euclidean average over the token embeddings at the final encoding layer and produce $\boldsymbol{z}_{\text{enc}} = \text{mean}(\boldsymbol{h}_{\text{enc},1}^{L_{\text{enc}}}, \ldots, \boldsymbol{h}_{\text{enc},T_{\text{enc}}}^{L_{\text{enc}}})$, which is expected to store concise and salient information in the input prompt. Conversely, the second stage processes the decoder's internal hidden state, which is represented by all the tokens at every layer of the decoder, $\boldsymbol{h}_{\text{dec}}^\ell = (\boldsymbol{h}_{\text{dec},1}^\ell, \ldots, \boldsymbol{h}_{\text{dec},T_{\text{dec}}}^\ell) \in \mathbb{R}^{d \times T_{\text{dec}}}, \ell = 1, \ldots, L_{\text{dec}}$. Since the decoder contains a large number of intermediate tokens (i.e., the sequence length $\times$ the number of decoder layers), employing a naïve architecture such as an MLP is expected to generate significant computational burden. To avoid this, we design a convolutional layer-based module to efficiently compress the tokens and extract the low-dimensional vector $\boldsymbol{z}_{\text{dec}}$. In particular, we apply a 1-d convolutional layer to aggregate the information across tokens at the same layer, treating each token element as a separate channel so that efficient compression can be achieved via multi-channel convolution. Following this step, 1-d max pooling is applied at the token dimension to halve the number of tokens, followed by 1-d adaptive max pooling to further reduce the number of tokens to 1. Through this process, we extract a single vector, $\boldsymbol{z}_{\text{dec}}^\ell \in \mathbb{R}^d, \ell = 1, \ldots, L_{\text{dec}}$ to represent each decoder layer. By concatenating $\boldsymbol{z}_{\text{dec}} = (\boldsymbol{z}_{\text{dec}}^1, \ldots, \boldsymbol{z}_{\text{dec}}^{L_{\text{dec}}})$ with $\boldsymbol{z}_{\text{enc}}$, the final hidden representation $\boldsymbol{z} = [\boldsymbol{z}_{\text{enc}}, \boldsymbol{z}_{\text{dec}}]$ is produced and fed to the hypernetwork.

**Output: Modulation vectors for HNNs**  As mentioned, the output of the hypernetwork serves to update the parameters of an HNN. However, inferring the entire set $\boldsymbol{\Theta}_{\text{HNN}}$ of model parameters, which is typically very high-dimensional, is computationally challenging. To mitigate this issue, we introduce a "modulation" technique for updating HNNs online, which allows the hypernetwork to infer only a low-dimensional latent vector. Considering Hamiltonian dynamics which depend on a number of a parameters $\boldsymbol{\mu}$ (such as mass, stiffness, etc.), the key idea of this modulation is to divide the HNN parameters into two parts: a base set of model parameters $\boldsymbol{\Theta}_{\text{base}}$ which are shared across all parameterized Hamiltonians $\mathcal{H}_\Theta^{(j)} \approx \mathcal{H}(\cdot, \boldsymbol{\mu}^{(j)})$, where $\boldsymbol{\mu}^{(j)}$ denotes the $j$-th sampled vector of system parameters, and an individual set of model parameters $\boldsymbol{\Theta}_{\text{indv}}^{(j)}$ which are specific to each parameterized Hamiltonian.

In this study, we model HNNs as multilayer perceptrons (MLPs) and leverage "shift modulation" (Sitzmann et al., 2020), a parameterization technique that adds a shift to the bias in each layer of an MLP. That is, the $\ell$-th layer of the modulated MLP can be defined as

$$\boldsymbol{h} \mapsto \sigma\left(\boldsymbol{W}^{(\ell)}\boldsymbol{h} + \boldsymbol{b}^{(\ell)} + \boldsymbol{s}^{(\ell,j)}\right),$$

where $\boldsymbol{\Theta}_{\text{base}} = \{(\boldsymbol{W}^{(\ell)}, \boldsymbol{b}^{(\ell)})\}$ are the shared base parameters while $\boldsymbol{\Theta}_{\text{indv}}^{(j)} = \{\boldsymbol{s}^{(\ell,j)}\}$ are individual parameters specific to the $j$-th Hamiltonian system. During training, the base model parameters are updated via standard gradient descent, while the individual parameters are inferred from the hypernetwork, i.e., $\boldsymbol{\Theta}_{\text{indv}}^{(j)} = \boldsymbol{f}^{\text{hyper}}(\boldsymbol{z}_j)$, where $\boldsymbol{z}_j$ is the latent vector associated to the Transformer input $\boldsymbol{Z}_j$. Given that the decoder predicts future states while the hypernetwork infers individual parameters for the HNN, we use the HNN $\mathcal{H}_\Theta$ to align the decoder's predictions with a valid symplectic flow.

To this end, once the Transformer NO has been queried, the predicted states $\tilde{\boldsymbol{u}}_j^{n_j}$ in the output $\tilde{\boldsymbol{y}}_j^{n_j}$ are fed as input to the HNN $\mathcal{H}_\Theta^{(j)}$, which is then used to compute symplectic gradients at these states. Integrating these symplectic gradients in time (with a conservative time integration scheme) then produces a symplectic flow in accordance with Hamilton's least-action principle. Here, we introduce the sympleciticty loss (defined in Eq. (3)) to guide the decoder's predictions in alignment with the trajectories computed using the symplectic gradients.

## 3.4 DECODER OUTPUT FORMATTING: DISCRETIZATION-BASED OUTPUT REPRESENTATION

In the recent literature on time-series foundation models, an effective strategy has been to discretize continuous target values through quantization and reformulate forecasting as a classification problem rather than direct regression (Stewart et al., 2023). This approach, as adopted for example in Chronos (Ansari et al., 2024), has been shown to improve predictive capability and stability of forecasting. Motivated by this, we also adopt this strategy in our work.

Among the output of the base Transformer model, $\tilde{\boldsymbol{y}}_j^{n_j} = \boldsymbol{f}^{\text{TF}}(\boldsymbol{t}^{n_j}, \boldsymbol{Z}_j)$, the target variables of the predicted states $\tilde{\boldsymbol{u}}_j^{n_j}$ are discretized by using a linear bucket strategy: with a generic variable $v$, given the global min–max range $[v_{\min}, v_{\max}]$ across train/val/test sets, we partition it into $K$ uniform bins $I_k = [v_{\min} + (k-1)\Delta, \ v_{\min} + k\Delta], \quad \Delta = \frac{v_{\max} - v_{\min}}{K}$. Each target $v$ is mapped to its bin index $c$, and we design the base model to produce logits, instead of real numbers, and train the model with cross-entropy loss on the predicted logits, $w \in \mathbb{R}^K$. At inference, we apply soft decoding, which recovers the predicted values, by computing $\hat{y} = \sum_{k=1}^K p_k \mu_k, \quad p_k = \text{softmax}(w)_k$, where $\mu_k$ is the midpoint of bin $I_k$. All together, the base Transformer model first predict logits, $\tilde{\boldsymbol{w}}_j^{n_j}$, and states $\tilde{\boldsymbol{u}}_j^{n_j}$ via soft-decoding.

### 3.5 TRAINING ALGORITHM

We now describe the training algorithm for AI-Hamilton, which comprises three main components: an encoder-decoder transformer, a dynamics extractor, a hypernetwork, and an HNN. All components are trained in tandem using standard stochastic gradient descent methods. However, because the HNN depends on the decoder's predictions as input, jointly training all components from the beginning often leads to poor performance—early in training, the decoder produces inaccurate future state predictions, which misguide the HNN. To address this issue, we adopt a two-phase training strategy.

The first phase trains only the encoder-decoder transformer by minimizing the discrepancy between the decoder's prediction and the ground truth; since our approach considers the discretized decoder output representation, we utilize cross-entropy to minimize the discrepancy, $\mathcal{L}_{\text{CE}} = \sum_j \text{CE}(\boldsymbol{w}_j^{n_j}, \tilde{\boldsymbol{w}}_j^{n_j})$. Along with this loss, to ensure the predicted values remain numerically accurate after de-quantization (soft-decoding), we complete the CE loss with the mean-squared error (MSE) $\mathcal{L}_{\text{mse}} = \sum_j \|\boldsymbol{y}_j^{n_j} - \boldsymbol{f}^{\text{TF}}(\boldsymbol{Z}_j, \boldsymbol{t}^{n_j})\|_2^2$.

In the second phase, we introduce the hypernetwork and the HNN, minimizing the symplecticity loss $\mathcal{L}_{\text{symp}} = \sum_{i,j} \|\dot{\tilde{\boldsymbol{q}}}_{j,i}^{n_j} - \partial_{\boldsymbol{p}} \mathcal{H}_{\boldsymbol{\Theta}_{\text{HNN}}}(\tilde{\boldsymbol{q}}_{j,i}^{n_j}, \tilde{\boldsymbol{p}}_{j,i}^{n_j})\|_2^2 + \|\dot{\tilde{\boldsymbol{p}}}_{j,i}^{n_j} + \partial_{\boldsymbol{q}} \mathcal{H}_{\boldsymbol{\Theta}_{\text{HNN}}}(\tilde{\boldsymbol{q}}_{j,i}^{n_j}, \tilde{\boldsymbol{p}}_{j,i}^{n_j})\|_2^2$ along with the prediction losses, where $\tilde{\boldsymbol{q}}_j^{n_j} = [\tilde{\boldsymbol{q}}_{j,T+1}^{n_j}, \ldots, \tilde{\boldsymbol{q}}_{j,2T}^{n_j}]$ and $\tilde{\boldsymbol{p}}_j^{n_j} = [\tilde{\boldsymbol{p}}_{j,T+1}^{n_j}, \ldots, \tilde{\boldsymbol{p}}_{j,2T}^{n_j}]$ denote the predicted trajectory for the $j$-th system. We use a linear combination as the total training loss: $\mathcal{L} = \lambda_{\text{mse}} \mathcal{L}_{\text{mse}} + \lambda_{\text{CE}} \mathcal{L}_{\text{CE}} + \lambda_{\text{symp}} \mathcal{L}_{\text{symp}}$ with hyper-parameters $\lambda_{\text{mse}}, \lambda_{\text{CE}}, \lambda_{\text{symp}}$.

### 3.6 AI-HAMILTON VARIANT

Taking inspiration from Noether networks, we propose a variant of AI-Hamilton that uses the same architecture and training algorithm but is trained with a conservation-law-enforcing loss. In this variant, $\mathcal{H}_{\Theta}$ is treated as an arbitrary quantity that should remain invariant over time. This property is enforced by minimizing $\sum_{i,j} \|\overline{\mathcal{H}_{\Theta}(\tilde{\boldsymbol{q}}^{n_j}, \tilde{\boldsymbol{p}}^{n_j})} - \mathcal{H}_{\Theta}(\tilde{\boldsymbol{q}}_{j,i}^{n_j}, \tilde{\boldsymbol{p}}_{j,i}^{n_j})\|_2^2$, where $\overline{\mathcal{H}_{\Theta}(\tilde{\boldsymbol{q}}^{n_j}, \tilde{\boldsymbol{p}}^{n_j})} = \frac{1}{n_j} \sum_i \mathcal{H}_{\Theta}(\tilde{\boldsymbol{q}}_{j,i}^{n_j}, \tilde{\boldsymbol{p}}_{j,i}^{n_j})$. This loss encourages the model to learn an energy-like quantity that remains conserved over the simulation trajectory. We denote this variant by AI-Hamilton (Energy).

## 4 EXPERIMENTS

We now present the results of our experimental evaluation, which consists of two main parts. First, we apply the proposed models to a diverse set of Hamiltonian systems to assess their accuracy and generalization. Second, we evaluate the scalability of the proposed methods on high-dimensional systems. We refer the reader to Appendix **??** for detailed experimental settings and to Appendix **??** for additional ablation studies on the CNN-based compressor and shift modulation.

**Model architecture** The transformer has the encoder and decoder; both of them consists of three MHA blocks with four attention heads per block and three FF blocks with two linear layers: one expanding hidden dimension to $4d$, followed by a GELU activation, and another one shrinking dimension back to $d$. Both the hypernetwork and the HNN have three hidden layers with 100 neurons in each layer, followed by ReLU and Tanh, respectively.

**Data generation** For all experiments, we consider parameterized Hamiltonian systems and, for each Hamiltonian system, we uniformly sample system parameters from the specified ranges in (Table **??** in the Appendix). We sample trajectories by solving initial value problems for varying 100 initial conditions per each sampled parameter.

In previous methods testing ICL-based NOs (e.g., (Yang et al., 2023)), both training and test samples are drawn from within the same bounded reion of the parameter space. In this wor, to make the evaluation more challenging, we consider both semi-OOD and OOD settings. In the semi-OOD case, we partition the parameter space into a checkerboard pattern, where adjacent blocks alternate in color (white/black). Training and validation samples are drawn from one set of blocks (white), while test samples are collected from the interleaved blocks (black). For the OOD case, test samples are drawn from regions lying entirely outside the original bounding box used for training.

For all the systems, we sample 200 systems for training, 40 systems for validation, and 40 systems for semi-OOD or OOD testing. We set the maximum number of examples in $\mathcal{D}$ for one Hamiltonian system to five; when generating the data prompt, we randomly choose the number of samples in each prompt from $\{1, 2, 3, 4, 5\}$ At test time, the number of examples in each prompt is fixed to 3.

We implement our code with the PYTORCH and TORCHDIFFEQ (Chen, 2018). We leave all other essential details in the Appendix. For all experiments (unless otherwise specified), we repeat the experiments for 3 varying random seeds.

Table 1: The mean-squared error (MSE) over 8 Hamiltonian systems compared between the baselines: MAML, ICON and the proposed methods: AI-Hamilton (Energy) and AI-Hamilton. Since all ICL models see three examples at test time, we compare them with the 3-shot MAML counterpart. The best performance for each system is marked **bold**.

| System types | Meta Learning | In-Context Learning | | |
|---|---|---|---|---|
| | MAML-*3 shots* | ICON | AI-Hamilton (Energy) | AI-Hamilton |
| **Mass-spring** | $1.91 \times 10^1 \pm 7.86 \times 10^0$ | $1.31 \times 10^{-2} \pm 8.49 \times 10^{-3}$ | $6.28 \times 10^{-3} \pm 1.03 \times 10^{-3}$ | $\mathbf{6.27 \times 10^{-3}} \pm 1.61 \times 10^{-3}$ |
| **Pendulum** | $1.31 \times 10^1 \pm 2.77 \times 10^0$ | $1.53 \times 10^{-1} \pm 2.34 \times 10^{-2}$ | $2.06 \times 10^{-2} \pm 8.83 \times 10^{-3}$ | $\mathbf{1.99 \times 10^{-2}} \pm 1.30 \times 10^{-3}$ |
| **Duffing** | $5.66 \times 10^{-1} \pm 1.78 \times 10^{-1}$ | $4.14 \times 10^{-3} \pm 1.69 \times 10^{-3}$ | $3.43 \times 10^{-3} \pm 1.18 \times 10^{-3}$ | $\mathbf{2.56 \times 10^{-3}} \pm 5.67 \times 10^{-4}$ |
| **Hénon-Heiles** | $\mathbf{3.01 \times 10^{-4}} \pm 1.06 \times 10^{-5}$ | $5.66 \times 10^{-4} \pm 1.11 \times 10^{-4}$ | $4.53 \times 10^{-4} \pm 7.86 \times 10^{-5}$ | $5.49 \times 10^{-4} \pm 8.17 \times 10^{-5}$ |
| **Magnetic Mirror** | $4.42 \times 10^{-3} \pm 5.24 \times 10^{-4}$ | $4.56 \times 10^{-3} \pm 4.10 \times 10^{-4}$ | $2.05 \times 10^{-3} \pm 3.16 \times 10^{-4}$ | $\mathbf{1.92 \times 10^{-3}} \pm 3.82 \times 10^{-4}$ |
| **Double Pendulum** | $3.90 \times 10^{-2} \pm 3.10 \times 10^{-3}$ | $3.87 \times 10^{-4} \pm 7.89 \times 10^{-5}$ | $3.23 \times 10^{-4} \pm 3.89 \times 10^{-5}$ | $\mathbf{3.17 \times 10^{-4}} \pm 3.52 \times 10^{-5}$ |
| **SAM (nonsingular)** | $2.52 \times 10^1 \pm 2.47 \times 10^0$ | $9.62 \times 10^{-2} \pm 3.56 \times 10^{-2}$ | $4.66 \times 10^{-2} \pm 5.68 \times 10^{-3}$ | $\mathbf{4.59 \times 10^{-2}} \pm 6.80 \times 10^{-3}$ |
| **SAM (singular)** | $1.41 \times 10^1 \pm 1.01 \times 10^0$ | $3.98 \times 10^{-2} \pm 1.02 \times 10^{-2}$ | $3.26 \times 10^{-2} \pm 4.89 \times 10^{-3}$ | $\mathbf{2.72 \times 10^{-2}} \pm 4.87 \times 10^{-3}$ |

## 4.1 Experiments on diverse sets of Hamiltonian systems

We first evaluate the performance of AI-Hamilton across eight diverse Hamiltonian systems spanning classical mechanics, astrophysics, and plasma physics — including the mass-spring, pendulum, double pendulum, duffing oscillator, Hénon–Heiles (HH) (Hénon & Heiles, 1964), magnetic mirror (MM) (Efthymiopoulos et al., 2015), and swinging Atwood's machine (SAM) (Tufillaro et al., 1984).

**Results** The full test results evaluated on semi-OOD on the above listed Hamiltonian systems are presented in Table 1. As baselines for comparison, we consider an optimization-based meta-learning algorithm, model-agnostic meta-learning (Finn et al., 2017), combined with HNNs, denoted as MAML, and an ICL-based algorithm, ICON (Yang et al., 2023). As the ICL-based methods utilize on-average 2.5 examples, we consider 3-shot update in MAML. Table 1 shows that, across the most systems, ICL-based methods outperforms the MAML-based method over some orders-of-magnitudes. Among the ICL-based methods, the AI-Hamilton methods achieve the lower mean error, outperforming ICON, and AI-Hamilton produces the smallest error. The method is effective even on challenging dynamics. In SAM (singular), which exhibit stiff dynamics, AI-Hamilton significantly improves performance, reducing the errors from $3.26 \times 10^{-2}$ to $2.72 \times 10^{-2}$. This result demonstrates AI-Hamilton's ability to handle stiff or highly nonlinear systems. Moreover, AI-Hamilton also demonstrates robustness across all systems. The reduction in standard deviation indicates that its predictions are not only more accurate but also more stable.

## 4.2 Experiments on higher-dimensional Hamiltonian systems

We now further investigate whether the effectiveness of AI-Hamilton scales with the dimensionality of the benchmark problems. To this end, we construct synthetic 3D/5D/7D systems below. For $q \in \mathbb{R}^n$, $p \in \mathbb{R}^n$, we consider polynomial Hamiltonians of the form

$$H(q,p) = \underbrace{\frac{1}{2}\sum_{i=1}^n p_i^2}_{\text{kinetic}} + \underbrace{\sum_{i=1}^n a_i q_i^2}_{\text{quadratic}} + \underbrace{\sum_{i<j} C_{ij} q_i^2 q_j}_{\text{bilinear couplings}} + \underbrace{\sum_{i=1}^n b_i q_i^4}_{\text{quartic}},$$

where the quadratic terms control baseline stability, quartic terms introduce nonlinear stiffness, and bilinear couplings mediate cross-coordinate interactions that can induce complex dynamics. We let $n \in \{3, 5, 7\}$, fix $a_i = b_i = 1$, and sample only the coefficients $C_{ij}$. By fixing $a_i$ and $b_i$, we fix the shape and depth of the potential well, effectively "locking in" how wide and stiff the well is, independent of the coupling terms. For all $n$, we uniformly sample 200 sets of system parameters for the training set among the interval $[0.5, 1.5]$, 40 sets of system parameters for the ID set in the same range. We test the OOD generalization capability of ICL models over a setup: $C_{ij} \in [0.4, 0.5]$ with 40 sets of parameters sampled. Across all systems, initial generalized positions $\boldsymbol{q}$ and momenta $\boldsymbol{p}$ are sampled in $[0, 1]$.

**Results** Table 2 reports the MSE results for the polynomial Hamiltonian benchmarks, comparing ICON, AI-Hamilton (Energy) and AI-Hamilton across both ID and OOD test regimes. Across all settings, both AI-Hamilton variants consistently improve upon ICON, confirming the benefits of introducing the constraints. While he results show that the overall performance is decreasing as the dimension increases, the AI-Hamilton variants produce the improved results in most cases.

Table 2: Mean-squared error (MSE) on high-dimensional Hamiltonian systems for ICON, AI-Hamilton (Energy), and AI-Hamilton. The best performance for each system is shown in **bold**. Models are trained for 10k epochs on 3D systems, 20k epochs on 5D systems, and 30k epochs on 7D systems.

| System Types | | ICON | AI-Hamilton (Energy) | AI-Hamilton |
|---|---|---|---|---|
| **Polynomial-3D** | ID | $1.84 \times 10^{-3} \pm 1.07 \times 10^{-4}$ | $\mathbf{9.24 \times 10^{-4}} \pm 4.77 \times 10^{-5}$ | $1.18 \times 10^{-3} \pm 2.24 \times 10^{-5}$ |
| | OOD | $2.89 \times 10^{-2} \pm 3.59 \times 10^{-3}$ | $\mathbf{2.27 \times 10^{-2}} \pm 2.18 \times 10^{-3}$ | $2.73 \times 10^{-2} \pm 2.93 \times 10^{-3}$ |
| **Polynomial-5D** | ID | $1.78 \times 10^{-2} \pm 6.92 \times 10^{-4}$ | $\mathbf{1.52 \times 10^{-2}} \pm 1.15 \times 10^{-3}$ | $1.54 \times 10^{-2} \pm 9.11 \times 10^{-4}$ |
| | OOD | $5.08 \times 10^{0} \ \ \pm 2.53 \times 10^{-1}$ | $3.59 \times 10^{0} \pm 1.60 \times 10^{0}$ | $\mathbf{3.30 \times 10^{0}} \ \ \pm 1.39 \times 10^{0}$ |
| **Polynomial-7D** | ID | $4.52 \times 10^{-1} \ \ 3.22 \pm \times 10^{-2}$ | $4.27 \times 10^{-1} \ \ 4.15 \pm \times 10^{-2}$ | $\mathbf{4.26 \times 10^{-1}} \ \ 2.61 \pm \times 10^{-2}$ |
| | OOD | $5.41 \times 10^{1} \ \ 6.67 \pm \times 10^{0}$ | $\mathbf{2.51 \times 10^{1}} \ \ 5.86 \pm \times 10^{0}$ | $3.01 \times 10^{1} \ \ 6.74 \pm \times 10^{0}$ |

Expectedly, generalization under OOD tends to be more challenging than under strong coupling. While the AI-Hamilton variants outperform ICON in all cases, the OOD samples not only cause the distribution shift challenges, but also the numerical issues. We conjecture that this arises from the underlying physics: in the sampling regimes, the interactions lead to trajectories where the magnitudes of the numerical values of $q$ and $p$ ($p$ in particular) become larger. This in turn challenges the quantization scheme used in our classification-based prediction, as the fixed binning struggles to faithfully capture the wider range of values, thereby degrading accuracy. This also partially explains why AI-Hamilton (Energy) performs better than AI-Hamilton in most cases. AI-Hamilton's symplectic loss relies on the time derivative of $q$ and $p$, which are obtained from numerical computation. As the rates of change in these variables increase, the numerical approximation becomes less accurate, potentially introducing larger errors in the loss.

## 5 RELATED WORK

**Neural operators**   In recent years, neural operators have emerged as a powerful framework for enabling data-driven surrogate modeling of (partial/ordinary) differential equations (PDEs/ODEs). A notable early contribution in this area is DeepONet (Lu et al., 2019; Wang et al., 2021), which consists of dual network components—branch and trunk networks—that collaboratively approximate the target operator. In contrast, alternative strategies focus on modeling operator learning via integral kernel approximations, representatively, Fourier neural operators (Li et al., 2021) and variants (Wen et al., 2022; Tran et al., 2023; Li et al., 2023; Cao et al., 2024). Despite these recent advancements, most existing NOs are trained to learn a fixed operator corresponding to a specific data distribution and, consequently, generally lack the flexibility to adapt to new tasks or contexts without retraining.

**ICL-based NOs**   Recent advances have demonstrated the potential of ICL for solving PDEs and learning neural operators. ICON (Yang et al., 2023) first introduced ICL for operator learning, allowing a single transformer to generalize across ODE problems via prompting and without retraining. with further work showing Further work demonstrated generalization to unseen PDE families (Yang & Osher, 2024). Zebra (Serrano et al., 2024) uses autoregressive transformers to model PDE dynamics, conditioning on previous trajectories for flexible, prompt-based prediction. Data-efficient ICL for neural operators has also been explored by (Cole et al., 2024), showing that even linear transformers can provably solve linear elliptic PDEs using prompt examples.

**Hamiltonian dynamics learning**   HNNs (Greydanus et al., 2019) introduce a framework for learning conservative dynamical systems by modeling the Hamiltonian function with a neural network, ensuring energy conservation and reversible dynamics. Symplectic ODE-Nets (Zhong et al., 2020) extend this idea by integrating symplectic structure and external control into the neural architecture. SympNet (Chen et al., 2020) and Symplectic Recurrent Neural Networks (Chen et al., 2020) further improved long-term stability by combining learned Hamiltonians with symplectic multi-step integrators. In a similar vein, Lagrangian neural networks (Cranmer et al., 2020; Lutter et al., 2018) propose to learn the Lagrangian directly, thereby supporting a related class of systems. Recent work has generalized these principles to graph-based settings (Gruber et al., 2023).

## 6 CONCLUSION

We investigated leveraging in-context learning (ICL) capabilities to build structure-preserving neural operators for Hamiltonian systems. To enable ICL, we adopted a large language model-like transformer architecture and further introduced a hypernetwork to infer the Hamiltonian function, which is then used to align the transformer's predictions in accordance with Hamilton's least-action principle. We focused on parametric Hamiltonian systems, evaluating the proposed approach on eight challenging benchmarks. The proposed method consistently outperformed existing baselines, demonstrating that ICL can be used as an effective meta-learning strategy in modeling and preserving the dynamics of Hamiltonian systems.

## 7 ETHICS STATEMENT

This work focuses on developing machine learning models for physical systems, specifically combining neural operators with Hamiltonian structures to model dynamical behavior. Our research is purely computational, with applications in the physical sciences. We do not foresee any ethical concerns arising from this work. There are no direct societal or human subjects implications associated with the methods or experiments presented.

## 8 REPRODUCIBILITY STATEMENT

We have provided all relevant details necessary to reproduce the results presented in this work, including descriptions of the datasets, model architectures, training procedures, and evaluation metrics. To further ensure reproducibility, we will make our code and configuration files publicly available upon acceptance of the paper.

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
