# 1 EXPERIMENTAL SETUP

This section provides details on the setup used in our experimentation. Table 1 reports the parameter ranges for all Hamiltonian systems considered.

Table 1: System parameters and sampling ranges for each Hamiltonian system. Parameters are sampled uniformly within the specified intervals. Following convention, for the Mass-Spring, Pendulum, Duffing, and Double Pendulum systems, the initial momentum is set to 0. Although $\lambda$ in Hénon-Heiles is typically fixed at 1, we sample it from the range $[0.5, 1.5]$. We also generalize the parameter range for the Magnetic Mirror system. The equations of motion for each system, along with the rationale behind the parameter ranges, are provided in Appendix 2.

| System | Parameters | $\in$ Range |
|---|---|---|
| Mass-Spring | $(m, k, q, p)$ | $\in [0.5, 5.5] \times [5, 10] \times [1, 2] \times \{0\}$ |
| Pendulum | $(m, l, q, p)$ | $\in [0.5, 1.5] \times [0.5, 1.5] \times [1, 2] \times \{0\}$ |
| Duffing | $(\alpha, \beta, q, p)$ | $\in [1, 6] \times [-6, -1] \times [0.5, 1.5] \times \{0\}$ |
| Hénon-Heiles | $(\lambda, x_0, y_0, p_{x_0}, p_{y_0})$ | $\in [0.5, 1.5] \times [-0.2, 0.2]^4$ |
| Magnetic Mirror | $(B_0, \beta_1, \rho_0, z_0, p_{\rho_0}, p_{z_0})$ | $\in [1.5, 2.5] \times [0.5, 1.5] \times [-0.5, 0.5]^4$ |
| Double Pendulum | $(m_1, m_2, l_1, l_2, G, a_1, a_2, p_{a_1}, p_{a_2})$ | $\in [0.5, 1]^2 \times [1, 2]^2 \times [0.5, \pi/2]^2 \times \{0\}^2$ |
| SAM (nonsingular) | $(M, m, G, r, \theta, p_r, p_\theta)$ | $\in [1.1, 9.9] \times \{1\} \times \{\pi/2\} \times \{0\} \times \{0\}$ |
| SAM (singular) | $(M, m, G, r, \theta, p_r, p_\theta)$ | $\in [1.1, 9.9] \times \{0.0001\} \times \{\pi/2\} \times \{12\} \times \{0\}$ |
| Polynomial (In-Distribution) | $(C_{ij}, a_i, b_i)$ | $\in [0.5, 1.5] \times \{1\} \times \{1\}$ |
| Polynomial (Weak Coupling) | $(C_{ij}, a_i, b_i)$ | $\in [0.4, 0.5] \times \{1\} \times \{1\}$ |

**Dataset generation.** For each type of Hamiltonian system, we uniformly sample 200/40/40 (train/validation/test) instances of system parameters from the specified ranges in (Table 1). For the low-dimensional systems, we follow the semi-OOD sampling scheme, where the specified range is divided into two interleaved and non-overlapping regions. The training and validation sets are sampled from the first region, while the test set are sampled from the other. For the polynomial systems, we follow the OOD sampling scheme. For each parameter instance, we sample 100 different initial conditions and generate trajectories consisting of 80 temporally equidistant measurements with a sampling frequency of 20 Hz (i.e., 20 measurements per second, 80 measurements in 4 seconds). When solving initial value problems (IVPs), we use the ground truth Hamiltonian to compute an accurate symplectic gradient at each point in phase space, and apply the Dormand–Prince method (DOPRI5) for time integration. We use the DOPRI5 implementation in PYTORCH TORCHDIFFEQ with absolute and relative tolerances set to $10^{-12}$.

**Network specifications.** The transformer configuration is identical across all models: both the encoder and decoder contain three multi-head attention (MHA) blocks, each with four attention heads. There are also three feed-forward (FF) blocks, each consisting of two linear layers: the first expands the embedding dimension to $4d$, followed by a GELU activation, and the second projects it back to dimension $d$. Both the hypernetwork and the HNN have three hidden layers with 100 neurons each. The hypernetwork uses ReLU activations, while the HNN uses Tanh. For fair comparison, the MAML model is implemented with the same dimensionality as the HNN. Since the ICL models receive three examples at test time, we set the number of inner steps in MAML to three, and the number of outer steps matches the number of training epochs used for the ICL models.

**Other implementation details.** We set the maximum number of examples in $\mathcal{D}$ for each Hamiltonian system to five. When generating the data prompt, we randomly sample the number of examples from $\{1, 2, 3, 4, 5\}$, so that the number of demonstrations per prompt is randomized during training. All experiments are run with three random seeds, and results are reported as the mean $\pm$ standard deviation. MAML experiments are conducted on Apple M4 Max chips (ARM architecture), while all other experiments are run on NVIDIA A100 GPUs (Ampere architecture, 80 GB memory). On average, training the ICL-based models for 10k epochs takes about one hour, while training MAML requires roughly four hours.

## 2 BENCHMARK SYSTEMS

In this section, we introduce the Hamiltonian functions of all considered systems.

**Spring-mass.**  The Hamiltonian function of the spring mass system is defined as:

$$\mathcal{H}(q, p; m, k) = \frac{p^2}{2m} + \frac{k(q - q_0)^2}{2}.$$

In a spring-mass system, $q$ represents the position of the mass, $p$ is the momentum, $m$ is the mass, and $k$ is the spring constant. The potential energy term $\frac{k(q-q_0)^2}{2}$ represents the restoring force of the spring when displaced from the equilibrium position $q_0 = 0$.

**Pendulum.**  The Hamiltonian function of the pendulum system is defined as:

$$\mathcal{H}(q, p; m, l, g) = \frac{p^2}{2ml^2} + mgl(1 - \cos(q - q_0)).$$

In a pendulum system, $q$ is the angular displacement, $p$ is the angular momentum, $m$ is the mass, $l$ is the length of the pendulum, and $g$ is the gravitational acceleration. The potential energy term $mgl(1 - \cos(q - q_0))$ accounts for the gravitational potential energy of the pendulum displaced from the vertical position $q_0 = 0$.

**Duffing Oscillator**  The Hamiltonian function of the Duffing oscillator is defined as:

$$\mathcal{H}(q, p; \alpha, \beta) = \frac{p^2}{2} + \frac{\alpha q^4}{4} + \frac{\beta q^2}{2}.$$

In the Duffing oscillator, $\alpha$ determines the strength of the nonlinear restoring force while $\beta$ represents the coefficient of the linear restoring force.

**Hénon–Heiles**  Hénon–Heiles gives the chaotic dynamics of a star around a galactic center with its motion constrained on a 2D plane. The Hamiltonian function is defined as:

$$\mathcal{H}(x, y, p_x, p_y; \lambda) = \frac{1}{2}\left(p_x^2 + p_y^2\right) + \frac{1}{2}\left(x^2 + y^2\right) + \lambda\left(x^2 y - \frac{y^3}{3}\right).$$

Though conventionally $\lambda$ is taken as unity, we sample it from the range $[0.5, 1.5]$ to increase system diversity.

**Magnetic-Mirror**  This system is a type of "Magnetic bottle" Hamiltonian system that describes the motion of a charged particle in an axisymmetric magnetic bottle. The Hamiltonian function is described as

$$H(\rho, z, p_\rho, p_z) = \tfrac{1}{2}\left(p_\rho^2 + p_z^2\right) + V(\rho, z; B_0, \beta_1),$$

with the parameterized potential

$$V(\rho, z; B_0, \beta_1) = \frac{B_0^2}{8}\rho^2 + \frac{B_0^2 \beta_1}{8}\rho^2 z^2 - \frac{B_0^2 \beta_1}{32}\rho^4 + \frac{B_0^2 \beta_1^2}{32}\rho^2 z^4 - \frac{B_0^2 \beta_1^2}{64}\rho^4 z^2 + \frac{B_0^2 \beta_1^2}{512}\rho^6.$$

This term plays the role of an effective potential coupling the radial $\rho$ and axial $z$ coordinates. When fixing $B_0 = 2, \beta_1 = 1$, we recover the equation:

$$\mathcal{H}(\rho, z) = \frac{1}{2}\left(\dot{\rho}^2 + \dot{z}^2\right) + \frac{1}{2}\rho^2 + \frac{1}{2}\rho^2 z^2 - \frac{1}{8}\rho^4 + \frac{1}{8}\rho^2 z^4 - \frac{1}{16}\rho^4 z^2 + \frac{1}{128}\rho^6.$$

**Double Pendulum**  The double pendulum is a classical mechanical system consisting of two pendula attached end-to-end, so that the motion of the second mass is influenced both by its own dynamics and the motion of the first. In the canonical Hamiltonian formulation used below, the system is described by two generalized coordinates $(a_1, a_2)$, representing the angles of the first and second pendula from the vertical, as well as their conjugate momenta $(p_{a1}, p_{a2})$. The Hamiltonian $\mathcal{H}$ captures both the kinetic and potential energy of the system and is given by the following:

$$\mathcal{H} = \frac{m_2 l_2^2 p_{a1}^2 + (m_1 + m_2)l_1^2 p_{a2}^2 - 2m_2 l_1 l_2 p_{a1} p_{a2} \cos(a_1 - a_2)}{2m_2 l_1^2 l_2^2 \left(m_1 + m_2 \sin^2(a_1 - a_2)\right)}$$
$$- (m_1 + m_2)gl_1 \cos(a_1) - m_2 gl_2 \cos(a_2).$$

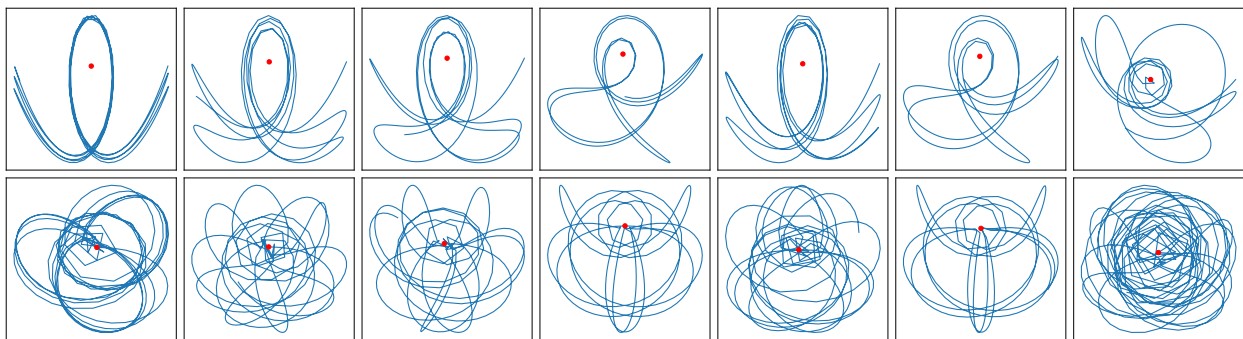

Figure 2: Example trajectories of nonsingular (top) and singular (bottom) SAMs. Each plot shows the Cartesian path of a swinging mass, with the origin marked in red.

**Swinging Atwood's Machine (SAM)**   The swinging Atwood's machine models the coupled dynamics of two masses connected by a string over a pulley, where one mass swings as a pendulum while the other moves vertically. The Hamiltonian is defined as:

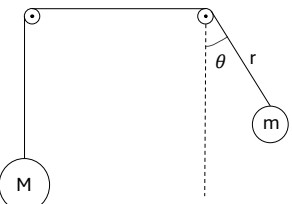

$$\mathcal{H}(r, \theta, p_r, p_\theta; m, M, g) = \frac{p_r^2}{2(M+m)} + \frac{p_\theta^2}{2mr^2} + gr(M - m\cos\theta).$$

Figure 1: Swinging Atwood's machine.

Following the setup studied by (**?**), we separate the SAM systems into two types, even though both share the same parameter range, $M \in [1.1, 9.9]$. Figure 2 depicts example trajectories of nonsingular and singular SAMs. In nonsingular SAMs, the swinging mass does not pass through the center, and the trajectory exhibits smooth turns with moderate acceleration. On the other hand, in singular SAMs, the mass passes through the origin, and the resulting trajectories often exhibit sharp transitions (i.e., rapid transitions in the positions and momenta) which tend to generate more diverse trajectory patterns.

**Polynomial Hamiltonian Systems**   We revisit the synthetic polynomial Hamiltonian systems proposed to evaluate performance on high-dimensional settings. For $q \in \mathbb{R}^n$, $p \in \mathbb{R}^n$, we consider polynomial Hamiltonians of the form

$$H(q, p) = \underbrace{\frac{1}{2}\sum_{i=1}^{n} p_i^2}_{\text{kinetic}} + \underbrace{\sum_{i=1}^{n} a_i q_i^2}_{\text{quadratic}} + \underbrace{\sum_{i<j} C_{ij} q_i^2 q_j}_{\text{bilinear couplings}} + \underbrace{\sum_{i=1}^{n} b_i q_i^4}_{\text{quartic}},$$

where the quadratic terms control baseline stability, quartic terms introduce nonlinear stiffness, and bilinear couplings mediate cross-coordinate interactions that can induce complex dynamics.

# 3 TRAINING ALGORITHM

Algorithm 1 describes the proposed two-phase training procedure for AI-Hamilton.

---

**Algorithm 1** Two-phase training procedure for AI-Hamilton.

---

**Require:** Training data $\mathcal{D}$, the maximum number of epochs $n_{\max_1}$ for phase 1 and $n_{\max_2}$ for phase 2, the weighting factors $0 < \lambda_{\text{CE}}, \lambda_{\text{symp}} < 1$

Initialize the model parameters of $\boldsymbol{f}^{\text{TF}}$, $\boldsymbol{f}^{\text{e}}$, $\boldsymbol{f}^{\text{hyper}}$, and $\mathcal{H}_{\boldsymbol{\Theta}}$

$\triangleright$ Phase 1: Optimize the transformer only

**for** $t = 1$ to $n_{\max_1}$ **do**
    Draw a batch of data $\mathcal{B}$
    **for** each training sample $b \in \mathcal{B}$ **do**
        Predict trajectory: $\tilde{\boldsymbol{y}}_b, \tilde{\boldsymbol{w}}_b \leftarrow \boldsymbol{f}^{\text{TF}}(\boldsymbol{Z}_b, \boldsymbol{t}^{n_b})$
    Compute loss: $\mathcal{L}_{\text{fitting}} = (1 - \lambda_{\text{CE}})\mathcal{L}_{\text{mse}} + \lambda_{\text{CE}}\mathcal{L}_{\text{CE}} = \frac{(1-\lambda_{\text{CE}})}{|\mathcal{B}|}\sum_{b \in \mathcal{B}} \|\boldsymbol{y}_b - \tilde{\boldsymbol{y}}_b\|_2^2 + \lambda_{\text{CE}}\text{CE}(\boldsymbol{w}_b, \tilde{\boldsymbol{w}}_b)$
    Update parameters of $\boldsymbol{f}^{\text{TF}}$ via gradient descent on $\mathcal{L}_{\text{fitting}}$

$\triangleright$ Phase 2: Jointly optimize all networks

**for** $t = 1$ to $n_{\max_2}$ **do**
    Draw a batch of data $\mathcal{B}$
    **for** each training sample $b \in \mathcal{B}$ **do**
        Predict trajectory: $\tilde{\boldsymbol{y}}_b \leftarrow \boldsymbol{f}^{\text{TF}}(\boldsymbol{Z}_b, \boldsymbol{t}^{n_b})$
        Extract latent: $\boldsymbol{z}_b \leftarrow \boldsymbol{f}^e(\boldsymbol{h}_{\text{enc}}^{L_{\text{enc}}}, [\boldsymbol{h}_{\text{dec}}^1, \ldots, \boldsymbol{h}_{\text{dec}}^{L_{\text{dec}}}])$
        Infer individual model parameters of an HNN: $\boldsymbol{\Theta}_{\text{indv}}^{(b)} \leftarrow \boldsymbol{f}^{\text{hyper}}(\boldsymbol{z}_b)$
        Modulate Hamiltonian: $\boldsymbol{\mathcal{H}}_{\Theta}^{(b)} \leftarrow \mathcal{H}_{\boldsymbol{\Theta}_{\text{base}} \cup \boldsymbol{\Theta}_{\text{indv}}^{(b)}}$
    Compute loss: $\mathcal{L}_{\text{fitting}} = (1 - \lambda_{\text{CE}})\mathcal{L}_{\text{mse}} + \lambda_{\text{CE}}\mathcal{L}_{\text{CE}} = \frac{(1-\lambda_{\text{CE}})}{|\mathcal{B}|}\sum_{b \in \mathcal{B}} \|\boldsymbol{y}_b - \tilde{\boldsymbol{y}}_b\|_2^2 + \lambda_{\text{CE}}\text{CE}(\boldsymbol{w}_b, \tilde{\boldsymbol{w}}_b)$
    Compute symplectic loss:

$$\mathcal{L}_{\text{symp}} = \sum_{i,b} \|\dot{\boldsymbol{q}}_{b,i}^{n_b} - \partial_{\boldsymbol{p}}\mathcal{H}_{\boldsymbol{\Theta}_{\text{HNN}}}(\tilde{\boldsymbol{q}}_{b,i}^{n_b}, \tilde{\boldsymbol{p}}_{b,i}^{n_b})\|_2^2 + \|\dot{\boldsymbol{p}}_{b,i}^{n_b} + \partial_{\boldsymbol{q}}\mathcal{H}_{\boldsymbol{\Theta}_{\text{HNN}}}(\tilde{\boldsymbol{q}}_{b,i}^{n_b}, \tilde{\boldsymbol{p}}_{b,i}^{n_b})\|_2^2$$

    Compute joint loss: $\mathcal{L}^{(t)} = (1 - \lambda_{\text{symp}})\mathcal{L}_{\text{fitting}} + \lambda_{\text{symp}}\mathcal{L}_{\text{symp}}$
    Update all parameters via gradient descent on $\mathcal{L}^{(t)}$

---

# 4 ABLATION ON CNN MODULE AND SHIFT MODULATION

We conduct an ablation study on both singular and nonsingular SAM systems. We analyze the impact of the proposed CNN-based decoder representation extractor compared to naïve mean aggregation, and the effect of shift modulation compared to full modulation. As shown in Table 2, the CNN-based extractor consistently outperforms naïve mean aggregation across most cases. Although the full modulation scheme provides only marginal improvements, it significantly increases the number of model parameters, since many more HNN parameters must be inferred. Therefore, choosing shift modulation offers a better trade-off between performance and model complexity.

Table 2: With an HNN of shape $[4, 100, 100, 100, 1]$ and a hypernetwork whose last hidden layer has 100 neurons (three hidden layers of 100 neurons each), the parameter requirements differ significantly. For full modulation, the hypernetwork must generate all 20,801 HNN parameters, leading to an additional $100 \times 20{,}801$ parameters in its output layer. In contrast, for shift-only modulation, it only needs to generate one shift per neuron (301 in total), resulting in just $100 \times 301$ additional parameters.

| Case | Nonsingular | | Singular | |
|---|---|---|---|---|
| (CNN, Mod) | Mean | Std. Dev. | Mean | Std. Dev. |
| $(\times, \times)$ | $4.29 \times 10^{-2}$ | $4.17 \times 10^{-3}$ | $3.26 \times 10^{-2}$ | $5.63 \times 10^{-3}$ |
| $(\checkmark, \times)$ | $4.60 \times 10^{-2}$ | $4.86 \times 10^{-3}$ | $2.85 \times 10^{-2}$ | $5.67 \times 10^{-3}$ |
| $(\times, \checkmark)$ | $4.76 \times 10^{-2}$ | $9.56 \times 10^{-3}$ | $3.19 \times 10^{-2}$ | $7.98 \times 10^{-4}$ |
| $(\checkmark, \checkmark)$ | $4.59 \times 10^{-2}$ | $6.80 \times 10^{-3}$ | $2.72 \times 10^{-2}$ | $4.87 \times 10^{-3}$ |

## 5   THE USE OF LARGE LANGUAGE MODELS (LLMs)

In this paper, LLMs were used exclusively for correcting grammatical errors and improving the clarity of writing.