# OpenReview forum: "AI-Hamilton: Leveraging In-Context Learning for Modeling Hamiltonian Systems"
_ICLR.cc/2026/Conference — ICLR 2026 Conference Withdrawn Submission_

### Official Review · Reviewer_bgNV · 2025-10-26

**Soundness:** 2
**Presentation:** 1
**Contribution:** 2
**Rating:** 2
**Confidence:** 3

**Summary:**

This paper proposes AI-Hamilton, a framework that integrates in-context learning (ICL) and Hamiltonian Neural Networks (HNNs) for modeling dynamical systems governed by Hamiltonian mechanics. The model uses a Transformer-based neural operator to perform in-context adaptation, and a hypernetwork to modulate the parameters of an auxiliary HNN, ensuring that predicted trajectories preserve energy and symplectic structure. Experiments on several canonical Hamiltonian systems show that AI-Hamilton achieves lower MSE than baselines such as ICON and MAML, and exhibits promising generalization in both in-distribution and out-of-distribution (OOD) tests.

**Strengths:**

**Strengths**

1. Integrating a Transformer with a hypernetwork-driven HNN is a novel idea that connects meta-learning with physical informed neural network that can preserve the Hamiltonian structure.

2. Results are presented on a diverse range of Hamiltonian systems (low- to high-dimensional, linear to chaotic), showing consistent improvements over ICON and MAML baselines.

3. Physically grounded motivation: The method explicitly enforces symplecticity, which is crucial for stable long-term prediction in conservative systems.

**Weaknesses:**

**Weaknesses**

1. The proposed pipeline (Transformer + CNN extractor + hypernetwork + HNN) is quite elaborate compared to the benchmark tasks (e.g., pendulum, spring, Duffing), which are simple toy models. It is unclear whether such complexity is necessary for systems of this scale, or if similar performance could be achieved with simpler, lightweight models. The model should be tested on more complex or realistic scenarios such as protein structure folding and chaotic n-body motions.

2. The assumption of the model is too strong: it requires full-state observability, and all experiments assume access to both position $q$ and momentum $q$, which is impractical in realistic scenarios. In many real-world systems only partial observations (e.g., positions only recorded by sensors) are available. The model’s performance under partial observability remains untested, limiting its practical significance.

3. Both training and test data are clean, noise-free trajectories. Since HNN-based approaches are known to be sensitive to noise due to the use of numerical derivatives from trajectories, the absence of noise robustness experiments weakens the empirical validation.

**Questions:**

Q1 : How does AI-Hamilton perform under partial observation, e.g., when only $𝑞(t)$ is available? Could the model infer missing momentum states or predict accurately the position?

Q2: Have the authors tested the robustness to noise, such as adding Gaussian noise or attack noise to training data, which are common in real world tasks?

Q3: Since the training involves multiple Hamiltonian systems simultaneously, how can we ensure that the model has truly learned the underlying physical principles, rather than performing joint multi-task learning? In particular, can the trained model generalize to unseen Hamiltonian families with qualitatively different energy landscapes?

---

### Official Review · Reviewer_EG7s · 2025-10-30

**Soundness:** 3
**Presentation:** 3
**Contribution:** 3
**Rating:** 6
**Confidence:** 3

**Summary:**

This paper presents AI-Hamilton, a novel framework for learning the dynamics of Hamiltonian systems from observational data. The core problem it addresses is the dichotomy between flexible but physically-agnostic in-context learning (ICL) models (like Transformers) and physically-consistent but less adaptive structure-preserving models (like Hamiltonian Neural Networks, HNNs). The authors propose an elegant solution that synergizes these two paradigms. The architecture consists of an encoder-decoder Transformer for ICL, which processes example trajectories (a "prompt") to understand the context of a new system. Crucially, instead of using the Transformer's output directly, its internal decoder representations are fed into a hypernetwork. This hypernetwork then modulates the parameters of a separate HNN, effectively configuring a system-specific Hamiltonian on the fly. The final dynamics are generated by this modulated HNN, thus ensuring physical consistency (symplecticity and energy conservation). The model is trained with a dual objective, combining a standard prediction loss for the Transformer with a symplecticity loss that aligns the Transformer's predictions with the physically-valid flow from the HNN. The authors demonstrate through extensive experiments on a variety of Hamiltonian systems, including high-dimensional and out-of-distribution scenarios, that AI-Hamilton consistently outperforms strong baselines like MAML and the ICL-based ICON.

**Strengths:**

- Novel and Sound Conceptual Contribution: The central idea of using a powerful but "black-box" ICL model to parameterize a separate, "white-box" structure-preserving model is highly innovative and compelling. It offers a principled way to inject strong physical priors into the flexible ICL framework without compromising the expressiveness of the Transformer.

- Well-Designed Architecture: The implementation is thoughtful and non-trivial. The use of a hypernetwork for online modulation of the HNN is a clever mechanism for in-context parameter inference. The design choices, such as the extractor for decoder hidden states, the "shift modulation" technique for efficiency, and the two-phase training strategy, demonstrate a deep understanding of the practical challenges.

- Comprehensive Empirical Evaluation: The paper's claims are backed by rigorous experiments on a diverse set of nine benchmark Hamiltonian systems, ranging from simple to chaotic and high-dimensional. The inclusion of semi-OOD and OOD test settings provides a strong test of generalization, which is a key goal of ICL.

**Weaknesses:**

- Complexity and Computational Cost: The proposed architecture, while effective, is significantly more complex than the baselines. It involves a Transformer, a CNN-based extractor, a hypernetwork, and an HNN. This complexity likely translates to higher computational overhead and more hyperparameters to tune. A discussion or analysis of the computational costs (e.g., training time, inference time, parameter count) relative to the baselines would be a valuable addition to contextualize the performance gains.

- Justification for Discretization-Based Output: The choice to reformulate the regression task as a classification one (Section 3.4) is an interesting design choice borrowed from recent time-series models. However, its motivation and impact are not fully explored. The authors note its potential drawbacks in OOD settings. An ablation study or clearer justification for why this is preferred over standard regression for the Transformer's pre-training would strengthen the paper.

**Questions:**

- Regarding the discretization strategy: What was the primary motivation for choosing this classification-based approach over standard MSE regression for the Transformer's output? Was it observed to be more stable during the initial training phase? How sensitive is the model's performance to the choice of the number of bins, K?

- Regarding the hypernetwork's role: The paper employs "shift modulation," which only updates the biases of the HNN. Have the authors considered or experimented with modulating other parts of the HNN, for instance, learning low-rank updates to the weight matrices? Could this provide more expressive power for adapting the Hamiltonian function, and at what computational cost?

- Regarding the Energy variant: The superior performance of this variant in some OOD cases is very interesting. Does this suggest that for particularly stiff or chaotic systems where numerical derivatives are unreliable, a softer conservation law constraint is a more practical and robust choice, even if it doesn't enforce the full symplectic structure?

- Could you provide a brief comparison of the training and/or inference times for AI-Hamilton versus the ICON baseline on a representative benchmark system? This would help readers better understand the trade-offs between performance and computational complexity.

---

### Official Review · Reviewer_dLKu · 2025-10-30

**Soundness:** 1
**Presentation:** 1
**Contribution:** 2
**Rating:** 2
**Confidence:** 4

**Summary:**

The paper proposes a novel framework that combines in-context learning (ICL) with Hamiltonian neural networks (HNNs) to accurately model energy-conserving physical systems. The authors introduce an encoder-decoder Transformer that learns from trajectory prompts, a hypernetwork that dynamically adjust HNN parameters and a training scheme ensuring symplecticity and energy conservation. This framework allows the model to adapt to unseen dynamical systems without retraining, preserving Hamiltonian structure while improving prediction accuracy. Extensive experiments demonstrates that AI-Hamilton consistently outperforms both meta-learning (MAML) and ICON.

**Strengths:**

The paper presents the development of an in-context learning framework that learns forecasting operators consistent with Hamilton’s principle, ensuring energy conservation and symplectic structure preservation.

The proposed methods consistently outperform the baselines under the authors’ experimental settings and report results with mean and standard deviation across multiple runs.

Extensive experiments across diverse benchmark Hamiltonian systems demonstrate that the evaluation covers a broad range of physical dynamics.

**Weaknesses:**

**The discussion of related work and baseline settings is insufficient.** One of the paper’s main contributions is the development of an ICL-based meta-learning framework for Hamiltonian systems, with MAML and ICON chosen as baselines. However, the difference between MAML and in-context learning is not clearly articulated, particularly regarding their problem formulations, training procedures, and which parameters are updated during adaptation. Including a comparative figure, table, or concise algorithmic summary would greatly improve clarity. Furthermore, prior studies [1, 2] have already proposed meta-learning frameworks for Hamiltonian systems; the authors should explicitly discuss how their approach differs from these works or at least justify why MAML adequately represents that research line as a baseline.

**Base models for ICL with neural operators.** Although the authors extensively discuss neural operators in the introduction, preliminaries, and related works, the implementation relies solely on a standard Transformer backbone. They do not incorporate or compare with more advanced neural operator architectures such as Fourier Neural Operators [3] or DeepONet [4], which are known to efficiently and flexibly encode physical states. Evaluating these architectures, or at least discussing their potential impact, would strengthen the paper’s technical completeness and support the claimed generality of the proposed framework.

**Lack of qualitative analysis.** While quantitative results are provided, the paper lacks qualitative evaluations that help interpret model performance. It is difficult to assess how the predicted trajectories compare visually across methods. Including trajectory plots or phase portraits for representative systems would significantly strengthen the paper’s contributions and demonstrate the fidelity of the learned dynamics. For instance, in the polynomial-7D OOD setting, an MSE on the order of 10¹ alone is insufficient to determine whether the predicted trajectories accurately capture the underlying system behavior.

**Unclear and overloaded notation.** The paper uses many variables, which makes several sections difficult to follow. Some descriptions are ambiguous, for example, the explanation of how the base Transformer model first predicts logits (around line 323) lacks clarity. Simplifying the notation or providing a concise notation table would greatly improve readability.

**Missing references.** The manuscript contains multiple unresolved reference markers (“??”) where tables, or appendices should be linked. The authors should correct these cross-referencing errors to ensure the paper’s completeness and professionalism.

[1] Identifying physical law of Hamiltonian systems via meta-learning, ICLR, 2021.

[2] Towards cross domain generalization of Hamiltonian representation via meta learning, ICLR, 2024.

[3] Fourier Neural Operator for Parametric Partial Differential Equations, ICLR, 2021.

[4] DeepONet: Learning Nonlinear Operators for Identifying Differential Equations Based on the Universal Approximation Theorem of Operators, Nature Machine Intelligence, 2021.

**Questions:**

**Clarification on Figure 1.**
What do the query and prompt represent in Figure 1? Additionally, is it intentional that the modulated HNN shows a white circle inside blue circles, where the white circle denotes modulation and the blue circle represents the original parameters?

**Trajectories vs. canonical coordinates.**
What is the precise difference between trajectories and the canonical coordinates (q,p)? Are they ever used interchangeably, or do they always represent distinct quantities?

**Parameter range selection.**
Could the authors explain how the parameter ranges were chosen. For instance, in the pendulum system, qis typically the angular displacement, so why is it sampled from $[1,2]$  instead of $[-\pi,\pi]$ or $[-\pi/2,\pi/2]$? Also, are the momentum p always fixed values?

**Minor.**
There is minor typo in line 207:
$$
q(t_i) = [q(t_i), \ldots, q_n(t_i)] \rightarrow q(t_i) = [q_1(t_i), \ldots, q_n(t_i)].
$$

---

### Official Review · Reviewer_r51H · 2025-11-01

**Soundness:** 3
**Presentation:** 3
**Contribution:** 3
**Rating:** 4
**Confidence:** 3

**Summary:**

First, the submission appears to deviate substantially from the official ICLR LaTeX style files. In particular, the text block size appears to differ from the required one.

This paper introduces AI-Hamilton, a novel neural operator framework that combines the few-shot adaptability of in-context learning (ICL) with the structure-preserving principles of Hamiltonian mechanics. The architecture integrates an encoder-decoder Transformer with a hypernetwork that outputs modulated parameters for a Hamiltonian Neural Network (HNN), enforcing symplecticity and energy conservation during training. Extensive experiments demonstrate strong performance across a variety of low- and high-dimensional Hamiltonian systems, including chaotic and stiff dynamics. The approach outperforms both meta-learning (MAML) and prior ICL-based operator learning methods (ICON), particularly under out-of-distribution (OOD) settings.

**Strengths:**

1. The integration of a Transformer-based ICL framework with a hypernetwork-modulated HNN is novel and addresses a critical gap between flexible learning and physics-consistent modeling.

2. Unlike prior ICL-based operator models (e.g., ICON), AI-Hamilton explicitly enforces physical structure via Hamilton’s least-action principle, improving both interpretability and long-term prediction fidelity.

3. The method consistently outperforms MAML and ICON across diverse Hamiltonian systems, including high-dimensional and OOD test cases. The performance gains are significant in both accuracy and stability.

4. The model does not require knowledge of underlying physical parameters at test time, relying solely on few-shot trajectory data, which broadens its applicability to real-world black-box systems.

**Weaknesses:**

The structure-preserving property is not clear. The Hamiltonian parameters are inferred from a hypernetwork, which raises concerns about whether the resulting map preserves symplectic structure. A symplectic map requires its Jacobian to be a symplectic matrix, but the parameterization via the hypernetwork introduces additional terms in the Jacobian that may violate this condition. During inference, the final trajectory is generated by the Transformer decoder rather than the HNN. While training uses a symplecticity loss to align the Transformer output with HNN dynamics, the actual predictions are not guaranteed to preserve structure at test time.

Although the method is intended to preserve physical structure, the experiments focus mainly on short-horizon MSE. There is no analysis of long-term trajectory fidelity (e.g., energy conservation over time), which is critical for assessing stability in Hamiltonian systems.

**Questions:**

Please see weakness above

---

### Note · Authors · 2025-12-01

I have read and agree with the venue's withdrawal policy on behalf of myself and my co-authors.